# A Multi-Scale Simulation Study of Irradiation Swelling of Silicon Carbide

**DOI:** 10.3390/ma15093008

**Published:** 2022-04-21

**Authors:** Chunyu Yin, Baoqin Fu, Yongjun Jiao, Zhengang Duan, Lei Wu, Yu Zou, Shichao Liu

**Affiliations:** 1Science and Technology on Reactor System Design Technology Laboratory, Nuclear Power Institute of China, Chengdu 610213, China; yincy909@163.com (C.Y.); yongjunjiao_npic@163.com (Y.J.); zgduan96@163.com (Z.D.); npicwulei@163.com (L.W.); hit_lsc@163.com (S.L.); 2Key Laboratory of Radiation Physics and Technology of the Ministry of Education, Institute of Nuclear Science and Technology, Sichuan University, Chengdu 610064, China; zouyu@scu.edu.cn

**Keywords:** silicon carbide, molecular dynamics, irradiation swelling, radiation damage, rate theory

## Abstract

Silicon carbide (SiC) is a promising structural and cladding material for accident tolerant fuel cladding of nuclear reactor due to its excellent properties. However, when exposed to severe environments (e.g., during neutron irradiation), lattice defects are created in amounts significantly greater than normal concentrations. Then, a series of radiation damage behaviors (e.g., radiation swelling) appear. Accurate understanding of radiation damage of nuclear materials is the key to the design of new fuel cladding materials. Multi-scale computational simulations are often required to understand the physical mechanism of radiation damage. In this work, the effect of neutron irradiation on the volume swelling of cubic-SiC film with 0.3 mm was studied by using the combination of molecular dynamics (MD) and rate theory (RT). It was found that for C-vacancy (C_V_), C-interstitial (C_I_), Si-vacancy (Si_V_), Si-interstitial (Si_I_), and Si-antisite (Si_C_), the volume of supercell increases linearly with the increase of concentration of these defects, while the volume of supercell decreases linearly with the increase of defect concentration for C-antisite (C_Si_). Furthermore, according to the neutron spectrum of a certain reactor, one RT model was constructed to simulate the evolution of point defect under neutron irradiation. Then, the relationship between the volume swelling and the dose of neutrons can be obtained through the results of MD and RT. It was found that swelling typically increases logarithmically with radiation dose and saturates at relatively low doses, and that the critical dose for abrupt transition of volume is consistent with the available experimental data, which indicates that the rate theory model can effectively describe the radiation damage evolution process of SiC. This work not only presents a systematic study on the relationship between various point defect and excess volume, but also gives a good example of multi-scale modelling through coupling the results of binary collision, MD and RT methods, etc., regardless of the multi-scale modelling only focus on the evolution of primary point defects.

## 1. Introduction

Silicon carbide (SiC) is a highly concerned covalent compound [1], since it has many important advantages as follows: high strength, high hardness, high corrosion resistance, high resistance to neutron irradiation activation and good thermal stability [2,3,4,5]. There exist hundreds of polytypes (e.g., cubic 3*C*-SiC (β-SiC), hexagonal 2*H*-SiC, 4*H*-SiC and 6*H*-SiC), while 3*C*-SiC is mostly used for structural application in nuclear reactors. Furthermore, the SiC ceramic composites show high specific strength, strong resistance to neutron irradiation and pseudo ductile fracture behavior, and SiC has great potential use in nuclear technology [6,7]. In light water reactor (LWR), SiC is considered as one of candidate materials for accident tolerant fuel cladding. In high temperature gas cooled reactor, SiC had been used as the structural “cladding” for TRISO coating fuel system [8]. In the 1950s, SiC coating was prepared for the nuclear fuel system of gas cooled reactor by chemical vapor deposition (CVD) [9]. SiC composites are also considered as candidate materials for fusion cladding in the magnetic confinement and inertial confinement fusion devices [10]. In addition, SiC is also a potential semiconductor material used in the field of high temperature, high power, high frequency, and high irradiation. Therefore, understanding the behavior of SiC under irradiation is wildly concerned nowadays.

The extreme service environment of nuclear technology (e.g., high temperature, high pressure and high flux fast neutron irradiation) will lead to degradation of the availability of SiC materials such as irradiated amorphous, irradiation swelling, irradiation creep, and irradiation fatigue [2,6,11,12,13]. Irradiation induced swelling has been widely studied for nuclear metal materials but few for SiC (some works up to 2007 were reviewed by Snead et al.) [2]. However, there is no doubt that irradiation swelling will become one of key issues for SiC servicing in nuclear reactor. Previous studies, conducted in a wide temperature range (900–1700 K) and under moderate/high flux neutron and Si irradiation, showed that the traditional transmission electron microscopy (TEM) was not enough to characterize the irradiation swelling. Leide et al. [12] utilized an ion implantation method to found that there exits large swelling-induced residual stress in ion implanted SiC. The irradiation swelling phenomenon is related to the generation and evolution of irradiation point defects (i.e., C-vacancy (C_V_), C-antisite (C_Si_), Si-vacancy (Si_V_), Si-antisite (Si_C_) and interstitial atoms of Si (Si_I_) and C (C_I_)). In the equilibrium state, the content of point defects is extremely low due to the high formation energy of point defects in SiC. But a large number of non-equilibrium point defects will be produced under irradiation, and different types of point defects have different generation rates. In 3*C*-SiC, the threshold energies of C and Si are 19 eV and 38 eV, respectively. Wang et al. [14] estimated the formation energy of intrinsic defects in 3*C*-SiC and their effect on electronic properties by ab initio method. Based on molecular dynamics (MD) studies, Devanathan et al. [15] and Ran et al. [16] show that the residual defects produced by primary knock-on atom (PKA) are mainly C-type Frenkel defects and anti-site defects. These point defects may annihilate each other (V + I) in the evolution process, form large-scale point defect clusters, or be absorbed by dislocations, dislocation loops, voids and grain boundaries. These processes were also affected by electronic stopping, which was not included in MD simulations [17]. It is generally believed that dislocation loops are easily formed at low temperature and voids are formed only at high temperature. According to the MD studies by Gao et al. [18] and Ran et al. [16], the cluster generation rate of residual defects is very low and the distribution is scattered. According to the results of Gao et al. [19], one high energy PKA can easily convert into multi-branch low energy PKAs, resulting in the formation of a number of small point defect clusters in inter-cascade.

Radiation damage of nuclear materials is one of key issues to design new fuel cladding materials. It is not enough to understand the radiation damage of nuclear materials just by the time-consuming and costly experimental studies. Thus, numerical simulation is required to understand the physical mechanism of the radiation damage. The details of the irradiation swelling usually involve multi-scale processes (e.g., the generation of point defects, the short-time-range interaction between these point defects, and the long-range migration and reaction of these point defects). Each process is studied with a different modeling tool and the prevailing approach to couple these multi-scale models (e.g., binary collision (BC) model, MD, and rate theory (RT), …) is through a hierarchical, information-passing paradigm [20]. In this work, MD and RT are used to study the microscopic physical mechanism of SiC volume swelling induced by fast neutron irradiation. Each type of point defects will induce strain in the lattice, and then lead to a certain volume change, causing swelling damage. The generation of point defects is modelling with BC and MD methods, the simply evolution of point defects is modelling with RT theory, and the relationship between various point defect and the excess volume (i.e., strain or swelling) is studied by MD simulation.

## 2. Model and Method

The research process includes the following steps. Firstly, MD is used to study the relationship between point defects (including concentration) and volume swelling (i.e., the relationship between structure and properties of materials under irradiation). Secondly, the number of elastic collisions (the number of PKA atoms) is estimated. Thirdly, the average energy of PKA produced by 1 MeV neutrons is calculated by GEANT4 software [21]. Based on the previous results calculated by MD simulations [16], the generation rate of primary point defects at the corresponding PKA energy is obtained. The relationship between neutron flux and the number of point defects is studied by using a rate theory model. The detailed process will be described later.

Potential selection is a key component of MD simulation. One Tersoff type potential [22] developed by P. Erhart et al. [23] is used in present work since it can effectively describe the mechanical and thermal properties of silicon, carbon and silicon carbide. The potential in the short-range distances was connected to the well-established Ziegler, Biesack, and Littmark (ZBL) potential [24] as previous work [16]. All of the MD simulations were performed using the LAMMPS software [25] developed by Sandia National Laboratory. The time step of MD simulation is fixed at 0.3 fs, based on the systematic convergence tests. And the size for the cubic supercell used in MD simulations is 20 *a* × 20 *a* × 20 *a* (as shown in Figure 1a), where *a* is the lattice constant of 3*C*-SiC, which is initialized to 4.36 Å. A simulation supercell size of 10 *a* × 10 *a* × 10 *a* was also used in all of the simulations discussed here and the main results and trends are exactly the same. Periodic boundary conditions are adopted in the three axes of the supercell. The perfect supercell contains 64,000 atoms (32,000 C + 32,000 Si) for the size of 20 *a* × 20 *a* × 20 *a*. Different numbers and types of point defects are introduced into the supercell to study the effect of point defects on the volume of the supercell. It should be noted that these point defects are randomly distributed throughout the supercell. 

The basic process of MD calculation is similar to that of previous studies [26,27]: the initial supercell with point defects is optimized to find the local lowest energy state, then the atomic velocity is set according to the Maxwell-Boltzmann distribution, and then 160,000 steps (48 ps) are run in the isothermal isobaric ensemble (NPT ensemble) based on MTK method [28] and PR method [29]. Two representative temperatures were considered in the simulations, i.e., 600 K (close to the operating temperature of fuel rod [30]) and 1500 K (as a high temperature), since the information of point defects at these temperatures produced by PKA has been studied in the previous work [16]. From 20,000 steps, the data are collected every 500 steps, and then some properties can be averaged to these time-saved data. For statistical accuracy, 16 independent simulations were performed under each combination of the given conditions.

## 3. Simulation Results and Discussion

### 3.1. Influence of Point Defects on the Volume of Supercell

Taking the effect of C-type point defects (including C_V_ and C_I_) on the volume of supercell at 600 K as an example. As shown in Figure 2, it can be seen that the aforementioned C-type defects cause the volume expansion, which means that the volume of supercell with the point defects is larger than that of perfect supercell (680,850.35 Å^3^ for the size of 20 *a* × 20 *a* × 20 *a* and 83,708.43 Å^3^ for the size of 10 *a* × 10 *a* × 10 *a*). It should be pointed out that the introduction of C vacancy in the supercell will also lead to an increase in the volume, while the volume of the metal system usually tends to decrease after the introduction of vacancy defect [31]. This is due to the fact that the metal atoms around vacancy shift to the vacancy center. However, SiC is a covalent bond system and each atom forms four covalent bonds with four neighbor atoms. When there is a vacancy in the SiC system, it means that one bond of the neighbor atom around the vacancy is destroyed, so the neighbor atom is dragged away by the other three bonds. Hence, the atoms around the vacancy are far away from the vacancy center (as denoted by the arrows in Figure 1b), and then the volume of the system becomes larger. The results are in agreement with previous findings of outward relaxation around vacancies in 4*H*-SiC [32,33].

The excess volume (*ν*) could be defined as the increased volume of the supercell induced by one additional point defect:*Ν* = (*V*_d_ − *V*_p_)/*N*_d_(1)
where *V*_d_ is the volume of the supercell with point defects, *V*_p_ is the volume of the perfect supercell and *N*_d_ is the number of point defects. 

The ratio (*R*_V_) of volume variation can be defined as, *R*_V_ = (*V*_d_ − *V*_p_)/*V*_p_. From Figure 2, it can also be found that the *R*_V_ increases linearly with the increase of the concentration (*c*_d_) of C-point defects, where *c*_d_ is *N*_d_/*N*_p_ and *N*_p_ is the number of lattice point of the perfect supercell, though the error bar (given by the standard deviation of the values) is noticeable for C_V_ (Figure 2a). The fitted *R*_V_~*c*_d_ relationship for C_V_ and C_I_ are as follows:*R*_V_ = −1.93 × 10^−6^ + 0.452*c*_d_(2)
*R*_V_ = −2.81 × 10^−7^ + 1.43C_I_(3)

The intercepts are both very close to zero when it is at 600. The excess volume of each point defect is related to the slope of each curve (*s*) in Figure 2.
*Ν* = *sV*_atom_,(4)
where *V*_atom_ = *V*_p_/*N*_p_ is the volume per atom in the perfect supercell. 

From the curves in Figure 2, it also shows that the excess volume caused by C_I_ is larger than that caused by C_V_. This phenomenon is similar to that in metal materials: the volume variation caused by interstitial atoms is also significantly larger than that caused by vacancy [31]. 

In addition, the excess volume of C-Frenkel pair (which is equivalent to the presence of both C_V_ and C_I_ in the supercell) was also studied as shown in Figure 2c. It can be found that the volume of the supercell also increases linearly with the addition of C-Frenkel pair, i.e., *R*_V_ increases linearly with the increase of the concentration (*c*_d_) of C-Frenkel. And the excess volume (19.54 Å^3^) of C-Frenkel pair is close to the sum of the excess volume caused by the corresponding number of C_V_ (6.12 Å^3^) and C_I_ (14.96 Å^3^), since the slope (1.87) of the *R*_V_~*c*_d_ curve of C-Frenkel is close to the sum of the slopes (0.452 + 1.43) of C_V_ and C_I_ as shown in Figure 2. 

As shown in Figure 3, the effect of Si-type point defects (including Si_V_, Si_I_ and Si-Frenkel) on the supercell volume at 600 K is similar to that of C-type point defects. The supercell volumes are also increasing linearly with the increasing of the number of point defects. The error bar is also noticeable for Si_V_ (Figure 3a), while the errors bar of Si_I_ (Figure 3b) and Si-Frenkel (Figure 3c) are smaller. The fitted *R*_V_~*c*_d_ relationship for Si_V_, Si_I_ and Si-Frenkel are described by the following expressions:*R*_V_ = −2.42 × 10^−6^ + 0.454*c*_d_(5)
*R*_V_ = −1.65 × 10^−6^ + 2.78*c*_d_(6)
*R*_V_ = 1.37 × 10^−6^ + 3.15*c*_d_(7)

The intercepts are also very close to zero when it is at 600 K. The slope of Si_I_ is larger than that of Si_V_, which means that the excess volume of Si_I_ is larger than that of Si_V_. Likewise, the excess volume (33.01 Å^3^) caused by Si-Frenkel pair is approximatively equal to the sum of excess volumes caused by Si_V_ (4.75 Å^3^) and Si_I_ (29.07 Å^3^), since the slope (3.15) of the *R*_V_~*c*_d_ curve of Si-Frenkel is close to the sum of the slopes (0.454 + 2.78) of Si_V_ and Si_I_ as shown in Figure 3.

In addition, MD simulations were also performed for the supercell containing anti-site point defects (C_Si_ and Si_C_), and the results at 600 K are shown in Figure 4. The volume of supercell with C_Si_ decreases with the increasing of the number of the point defects, which means that the introduction of C_Si_ will reduce the volume of the supercell. This is due to the formation of some C-C bonds. The bond length of C-C bond (*d*_C-C_ = 1.558 Å [34]) is shorter than that of C-Si bond (*d*_C-Si_ = 1.888 Å), so the volume of supercell with C_Si_ shrinks. For the Si_C_, since the bond length (*d*_Si-Si_ = 2.350 Å [34]) is longer than that of C-Si bond and some Si-Si bonds are formed due to the replacement of C by Si, the excess volume of Si_C_ is positive. Furthermore, there is also a linear relationship between the supercell volume and the number of the point defects. 

The fitted *R*_V_~*c*_d_ relationship for C_Si_ and Si_C_ are described by the following expressions:*R*_V_ = −9.59 × 10^−7^ − 1.1*c*_d_(8)
*R*_V_ = −8.87 × 10^−7^ + 1.31*c*_d_(9)

The intercepts are also very close to zero when it is at 600 K. It should be noted that the slopes have opposite sign but closed amount. These results are also in agreement with the earlier calculations [32,33] for 4*H*-SiC.

To analyze the temperature effects on the volume swelling, MD simulations for supercells with various point defects (similarly including C-type point defect, Si-type point defects and anti-site point defect) at 1500 K were also performed. The simulation results are shown in Figure 5, Figure 6 and Figure 7. Similar to that at 600 K, the introduction of C_V_ and Si_V_ can cause the volume of the supercell to expand; the excess volume of the interstitial atom is larger than that of the vacancy; the excess volume caused by C_Si_ atom is also negative; the volume of supercell with point defects decreases/increases linearly with the increasing of the number of point defects; the excess volume caused by Frenkel pair is approximatively equal to the sum of excess volume caused by the vacancy and the interstitial. 

The fitted *R*_V_~*c*_d_ relationships are described by the following expressions:*R*_V_ = −2.19 × 10^−6^ + 0.466*c*_d_ (C_V_)(10)
*R*_V_ = −1.22 × 10^−6^ + 1.51*c*_d_ (C_I_)(11)
*R*_V_ = 7.35 × 10^−7^ + 0.433*c*_d_ (Si_V_)(12)
*R*_V_ = −2.37 × 10^−7^ + 2.87*c*_d_ (Si_I_)(13)
*R*_V_ = 1.5 × 10^−7^ − 1.09*c*_d_ (C_Si_)(14)
*R*_V_ = −6.73 × 10^−7^ + 1.36*c*_d_ (Si_C_)(15)

The intercepts are also very close to zero when it is at 1500 K. The error bar is also noticeable for C_V_ (Figure 5a) and Si_V_ (Figure 6a), while the error bars of C_I_ (Figure 5b) and Si_I_ (Figure 6b) are smaller.

These aforementioned results can also be reproduced in the simulations of the small-size supercell (10 *a* × 10 *a* × 10 *a*). As described above, the excess volumes (*ν*) of various point defects at 600 K and 1500 K, which can be deduced from the slopes (*s*) of each curve in Figure 2, Figure 3, Figure 4, Figure 5, Figure 6 and Figure 7, *ν* = *sV*_atom_, are summarized in Table 1. It can be seen that the excess volume of Si_V_ is smaller than that of C_V_; while the excess volume of the Si_I_ is larger than that of the C_I_, and the excess volume of the interstitial is larger than that of the vacancy. The excess volume of C_Si_ is negative while that of Si_C_ is positive, but the absolute value is close. As shown in parentheses in Table 1, the results deduced from the simulations of the small-size supercell (10 *a* × 10 *a* × 10 *a*) are very close to the above results, which mean that the simulation size is large enough to obtain the convergent results. Besides, the temperature dependence on excess volume of point defects is not significant, thus the volume swelling at different temperatures is mainly dependent on the type, number and distribution of defects. It should be pointed out that the relation between excess volume and point defects is established in the case of low concentration. In the case of high concentration, point defects could interact with each other, then black spot defects and dislocation loops will certainly form, which may be inconsistent with the linear superposition rule. Thus, a more completed study should include the effects of defect clusters especially at high concentration of point defects. However, the concentration of point defects is low during the low dose neutron irradiation process. It should be noted that Xi et al. [35] had also studied the impact of point defects on the volumetric swelling of 3*C*-SiC at 300 K and 600 K with MD simulations of the small-size supercell (10 *a* × 10 *a* × 10 *a*) using the Tersoff potential fitted by Devanathan et al. [36]. Similar to this work, they also found that there exists an approximated linear relationship between the excess volume and the concentration of point defects and the vacancies have lower influence on the excess volume than the other point defect [35].

### 3.2. Effects of Neutron Dose on Volume Swelling

In fact, the effect of neutron irradiation on the swelling behavior of materials can be studied by analyzing the effect of irradiation-induced defects on the supercell volume. As described in the above, each type of point defects has a different excess volume, which can be regarded as almost independent of the concentration of point defects in the low concentration range. Therefore, the effect of neutron flux on volume swelling can be studied by estimating the type and concentration of point defects at different neutron flux.

In the first stage, the frequency of elastic collisions between the neutron flux and the SiC material in the nuclear reactor is estimated. Based on the standard provided by the American Society for testing and materials (ASTM) [37], the equivalent neutron flux of 1 MeV (*Φ*_ed, 1MeV_) is 1.92 × 10^12^ n/s/cm^2^. According to the results of Huang et al. [38], the average cross section (*σ*) of elastic collision is about 3.5 barns for SiC, and the cross section of C is twice that of Si. The atomic density (*N*) of SiC can be calculated to be 8/(4.359 × 10^−10^)^3^, so it can be estimated that the number of elastic collisions between SiC and neutrons is as follows:*ΦNσd* = 1.92 × 10^12^ × 8/(4.359 × 10^−10^)^3^ × 3.5 × 10^−28^ × 0.0003 = 1.947 × 10^10^ PKA/s/cm^2^,(16)
where the thickness (*d*) of SiC film service in the reactor is estimated to be 0.3 mm. 

These elastic collisions can be divided into 1.3 × 10^10^ PKA-C and 0.65 × 10^10^ PKA-Si. According to the calculation by GEANT4 software [21], the average energy of PKA-C produced by 1 MeV neutron is 1.5 × 10^5^ eV, and the average energy of PKA-Si is 5.6 × 10^4^ eV.

According to the previous work [16], production rates of different types of point defects are different. The relationship between the number of residual point defects (*N*_MD_) and PKA energy (*E*) can be described by the formula as follows:*N*_MD_ = *aE^b^*,(17)
where *a* and *b* are fitting parameters, and they are related to the type of point defects and temperature. 

Since the fitting parameter (*b*) is close to 1, the number of residual point defects increases almost linearly with the increasing of PKA energy, which means that *N*_MD_ is proportional to *E*. The number of point defects caused by 10 keV PKA is about 160 [16], then the *N*_MD_ of PKA-C and PKA-Si at the average energy can be deduced. Combined with Equation (16), so the generation rate of point defects can be estimated to be about 1.28 × 10^−8^ s^−1^. In light of the situation that the time scale of molecular dynamics calculation is limited, about 75% of the point defects will annihilate each other in the subsequent recovery stage of cascade process. It is also considered in previous research [39] that at least 60% of the adjacent Frenkel pairs will quickly recombine and annihilate. And it is assumed that the ratio of C and Si defect is 0.86 and 0.14, respectively. Then, the generation rates of different types of point defects are as follows:*K*_0_(C_I_/C_V_) = 1.38 × 10^−9^/s and *K*_0_ (Si_I_/Si_V_) = 0.22 × 10^−9^/s.(18)

The concentration of these point defects will not accumulate linearly with the increase of service time of SiC film, since these point defects will diffuse, be absorbed by various sinks, react with each other form various complex defect cluster, or annihilate each other. The time scale of these dynamic processes is much larger than that of MD calculation. Therefore, other calculation methods, such as Kinetic Monte Carlo (KMC) and rate theory, are required.

It is assumed that there is no sink in the interior of SiC film and only two interfaces of the film can be considered as the unbiased sinks for point defects. Therefore, the evolution of the concentration of the point defects in the SiC can be described by the following formula:(19)$$\frac{dC_{v}}{dt}=K_{0}-K_{\mathrm{iv}}C_{i}C_{v}+\nabla \cdot D_{v}\nabla C_{v},$$
(20)$$\frac{dC_{i}}{dt}=K_{0}-K_{\mathrm{iv}}C_{i}C_{v}+\nabla \cdot D_{i}\nabla C_{i},$$
where *C_v_* and *C_i_* are the concentrations of vacancy and interstitial atoms (atomic ratio), *K*_0_ is the average generation rate of point defects during irradiation, *K*_iv_ is the recombination rate of vacancy and interstitial atoms, *D_v_* and *D_i_* are the diffusion coefficients of vacancy and interstitial atoms, respectively. It should be pointed out that only the recombination events between vacancy and interstitial atoms of C and Si are taken into consideration in the calculation process, while the other events, including the diffusion of anti-site defects, are ignored.

According to the previous works [19,40], the capture radius of C-Frenkel pairs for self-recombination is 2.1 Å, and that of Si-Frenkel pairs is 6.3 Å; the diffusion coefficients of C_I_, Cv, Si_I_ and Siv are 1.23 × 10^−7^ exp (−1.76/*k*_b_*T*) m^2^/s, 7.26 × 10^−8^ exp (−3.661/*k*_b_*T*) m^2^/s, 3.3 × 10^−7^ exp (−0.829/*k*_b_*T*) m^2^/s and 7.26 × 10^−8^ exp (−2.4/*k*_b_*T*) m^2^/s, respectively.

To solve the above rate equations, the initial conditions and boundary conditions should be set. The initial condition is that the initial concentration of point defects in the SiC film is the concentration at thermodynamic equilibrium when SiC is not under irradiation (0 s). The formation energies of C_I_, Cv, Si_I_ and Siv are 6.95 eV, 4.19 eV, 8.75 eV and 4.97 eV, respectively. Then, the initial concentration of these point defects in the SiC can be calculated based on these formation energies. The defect concentration is approximately proportional to the Boltzmann factor (exp (−*E*_f_/*k*_b_*T*)) in the low concentration range. The boundary condition is that when all of the point defects migrate to the boundary (surface or interface between SiC film and substrate), all of the point defects disappear (are absorbed by the interface sinks), meaning that these interfaces are considered as the unbiased sinks for all of the considered point defects. The thickness of SiC film is set as 0.3 mm as mentioned above. In order to solve the above partial differential equations, the thickness direction of SiC film is divided into 1400 non-equidistant grids, in which the grids near to the boundary are dense and the grids in the middle are sparse. The initial rate theoretical simulation is carried out to solve the partial differential equation, and the time-dependent concentrations of different point defects in each grid of the SiC film can be obtained. Then, the swelling ratio (*ξ*) of the supercell can be obtained according to the defect concentration (*c*) in the SiC film and the excess volume (*ν*) obtained in the previous part can be expressed as:(21)$$\xi \left(t,x\right)=\frac{1}{V_{\mathrm{atom}}}\sum_{j=1}^{4}v_{j}c_{j}\left(t,x\right),$$
where *j* denotes four kinds of point defects: C_I_, Cv, Si_I_ and Siv; the anti-site defects are not considered in the calculation, and it should be note that the excess volumes of the two kinds of anti-site defects tend to counteract each other. The concentration *c*(*t*,*x*) of these point defects is a function of time (*t*) and position (*x*), and *ν* is the excess volume, and *V*_atom_ is the average monatomic volume of perfect SiC. The relationship between radiation swelling rate and equivalent neutron flux is shown in Figure 8 (600 K) and Figure 9 (1500 K). Two sets of excess volumes as shown in Table 1, from small-size supercell (10 *a* × 10 *a* × 10 *a*) simulations and large-size supercell (20 *a* × 20 *a* × 20 *a*) simulations, were used in the calculation, and the results are displayed as black curves and red curves in Figure 8 and Figure 9, respectively. It can be seen that the excess volume has little effect on the relationship between swelling ratio and radiation dose.

As shown in Figure 8 and Figure 9, the irradiation swelling slowly increases initially but significantly increases with radiation dose, and when reaching a critical neutron flux, the irradiation swelling then becomes stable. The critical neutron flux at 1500 K is higher than that at 600 K, which means that the higher the temperature is, the higher the critical neutron flux is. It is found that the critical neutron flux is 10^18^–10^24^ n/cm^2^. According to the two figures, the swelling rate is lower at higher temperature (1500 K), which is due to the increase of the mobility of point defects at higher temperature. The point defects are easier to recombine, annihilate and diffuse to the interface sinks, so fewer defects remain in the interior of the film. It should be noted that the formation of defect clusters, such as irradiated dislocation loops and voids, and the absorption and emission process of these defect clusters, are not considered in the rate theoretical model, although these events may also affect the swelling of the film. Miyazaki et al. [41] studied the neutron irradiation behaviors of β-SiC with neutron flux of 3.0 × 10^20^ to 1.7 × 10^23^ n/cm^2^ (*E* > 0.1 MeV). It was found that the critical neutron flux was 4.8 × 10^22^ n/cm^2^ at 370 to 620 °C, and the swelling was almost negligible when the irradiation temperature is higher than 1500 K. The swelling data between 1000 °C and 1200 °C are nearly not reported by snead et al. [2], while the fast neutron flux for irradiation saturation damage below 900 °C is 0.3–0.6 × 10^25^ n/m^2^ [42]. The isothermal relaxed experiments of α-SiC irradiated by neutrons at different temperatures [42] show that the swelling of SiC decreases linearly with the increase of temperature below 800 °C, which is consistent with the results of this work.

## 4. Conclusions

In this work, the effect of fast neutron irradiation on the volume swelling of SiC film is studied by means of molecular dynamics (MD) and rate theory (RT) methods. The main findings can be summarized as follows:(1)Based on the fast neutron energy spectrum, the evolution of irradiation swelling of SiC film along with fast neutron flux was simulated by the rate theory model ignoring the anti-site defects. It is found that the irradiation swelling rate first changes slowly, then increases significantly near one critical equivalent neutron flux, and finally tends to be constant with the increase of equivalent neutron flux. With the increase of temperature, the irradiation swelling rate decreases and the critical equivalent neutron flux increases, which is consistent with relevant literatures, indicating that the rate theoretical physical model applied in this study can effectively describe the evolution process of SiC irradiation damage.(2)It is found that for C_V_, Si_V_, Si_I_, C_I_ and Si_C_, the volume of supercell increases linearly with the increase of defect concentration: for C anti-site point defect (C_Si_), the supercell volume decreases linearly with the increase of defect concentration; the absolute value of the excess volume caused by Si anti-site point defect (Si_C_) is close to that by C_Si_; the excess volume of Si-vacancy (Si_V_) is smaller than that of C-vacancy (C_V_), while the excess volume of Si-interstitial (Si_I_) is larger than that of C-interstitial (C_I_); the excess volume induced by interstitial defects is larger than that of vacancy defects.(3)The excess volume increases linearly with the increase of defect concentration when the defect concentration is low.

## Figures and Tables

**Figure 1 materials-15-03008-f001:**
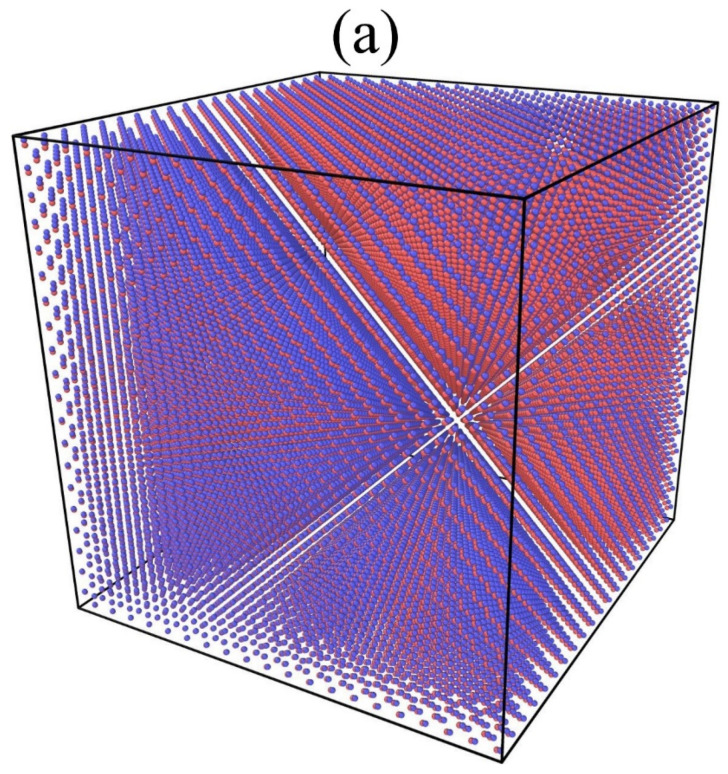
(**a**) The initial supercell without defects (the size is 20 *a* × 20 *a* × 20 *a*), (**b**) the structure near one C_V_. The blue sphere represents the C atom, the red sphere represents the Si atom and the deeppink sphere pointed by the arrow represents the first neighbor Si atom of C_V_, the directions of the arrows denote the offset of these first neighbor Si atom from the initial lattice position.

**Figure 2 materials-15-03008-f002:**
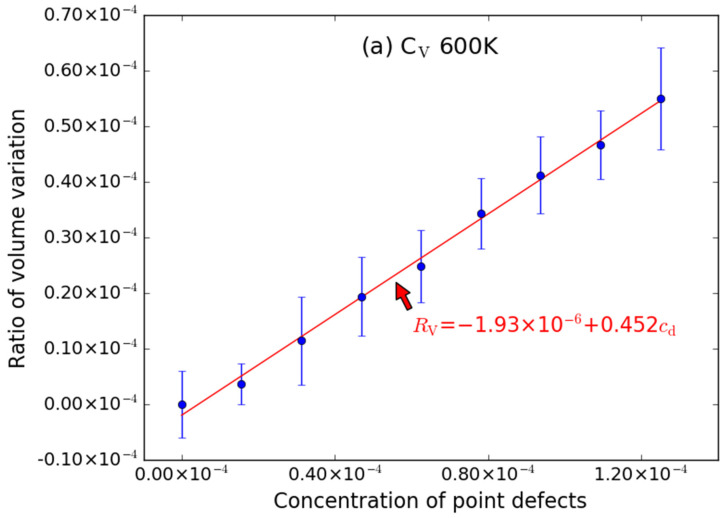
The ratio volume variation (*R*_V_) of supercell caused by C-type point defects ((**a**) C_V_, (**b**) C_I_ and (**c**) C-Frenkel pair) as function of concentration (*c*_d_) of point defects at 600 K. The error bars denote the standard deviations of the values. The size of supercell in MD simulation is 20 *a* × 20 *a* × 20 *a*.

**Figure 3 materials-15-03008-f003:**
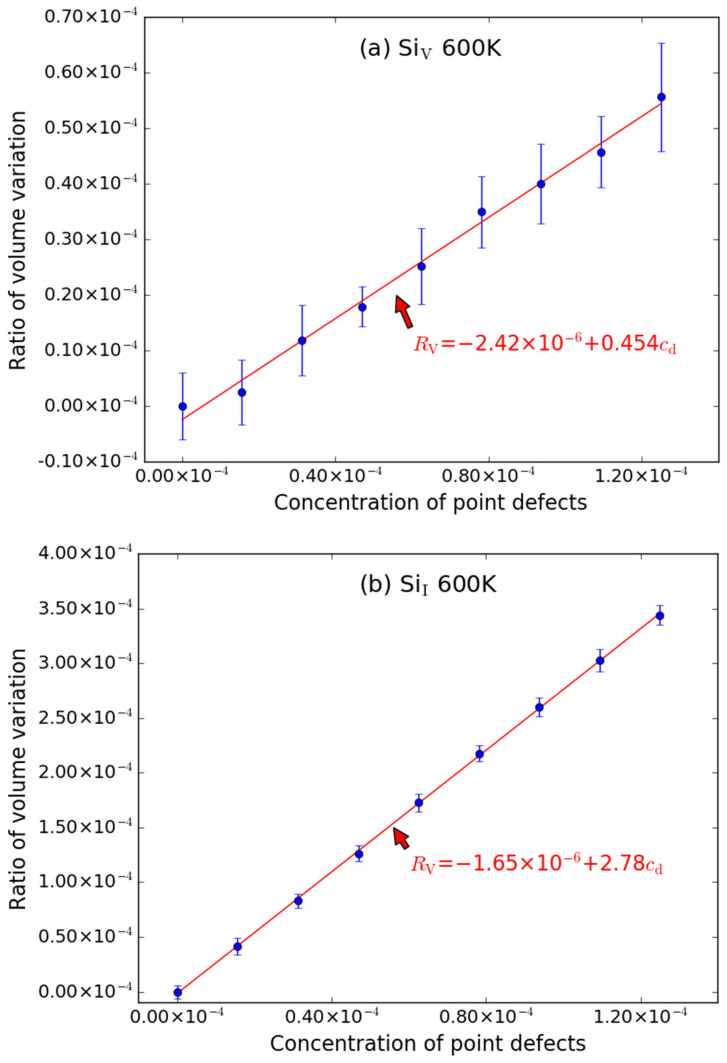
The ratio volume variation (*R*_V_) of supercell caused by Si-type point defects ((**a**) Si_V_, (**b**) Si_I_ and (**c**) Si-Frenkel pair) as function of concentrations (*c*_d_) of point defects at 600 K. The error bars denote the standard deviations of the values. The size of supercell in MD simulation is 20 *a* × 20 *a* × 20 *a*.

**Figure 4 materials-15-03008-f004:**
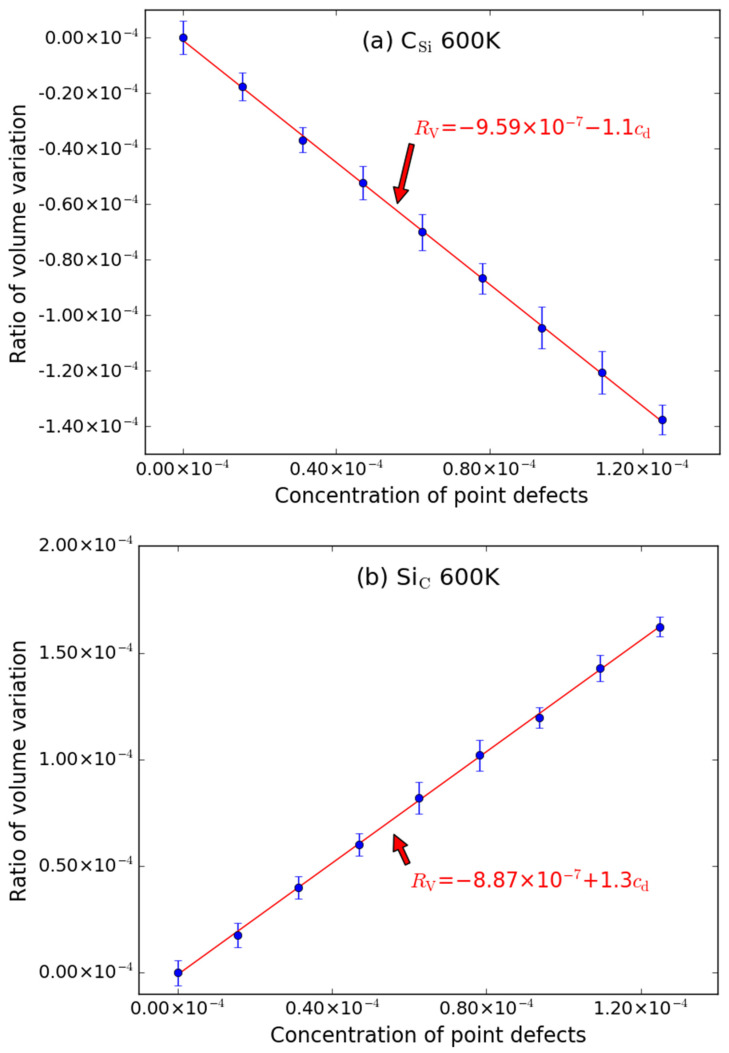
The ratio volume variation (*R*_V_) of the supercell caused by anti-site point defects ((**a**) C_Si_ and (**b**) Si_C_) as function of concentrations (*c*_d_) of point defects at 600 K. The error bars denote the standard deviations of the values. The size of supercell in MD simulation is 20 *a* × 20 *a* × 20 *a*.

**Figure 5 materials-15-03008-f005:**
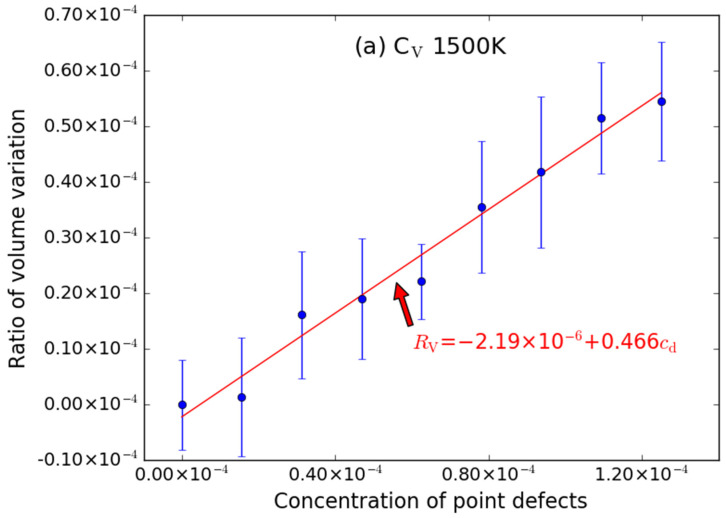
The ratio volume variation (*R*_V_) of the supercell caused by C-type point defects ((**a**) C_V_ and (**b**) C_I_) as function of concentrations (*c*_d_) of point defects at 1500 K. The error bars denote the standard deviations of the values. The size of supercell in MD simulation is 20 *a* × 20 *a* × 20 *a*.

**Figure 6 materials-15-03008-f006:**
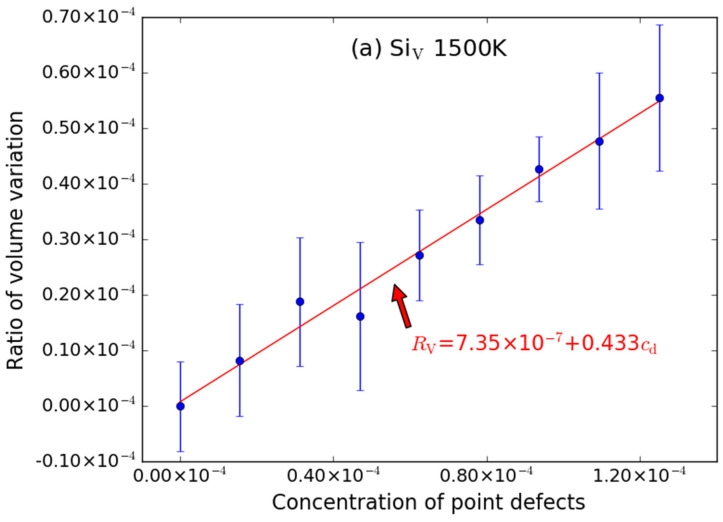
The ratio volume variation (*R*_V_) of the supercell caused by Si-type point defects ((**a**) Si_V_ and (**b**) Si_I_) as function of concentrations (*c*_d_) of point defects at 1500 K. The error bars denote the standard deviations of the values. The size of supercell in MD simulation is 20 *a* × 20 *a* × 20 *a*.

**Figure 7 materials-15-03008-f007:**
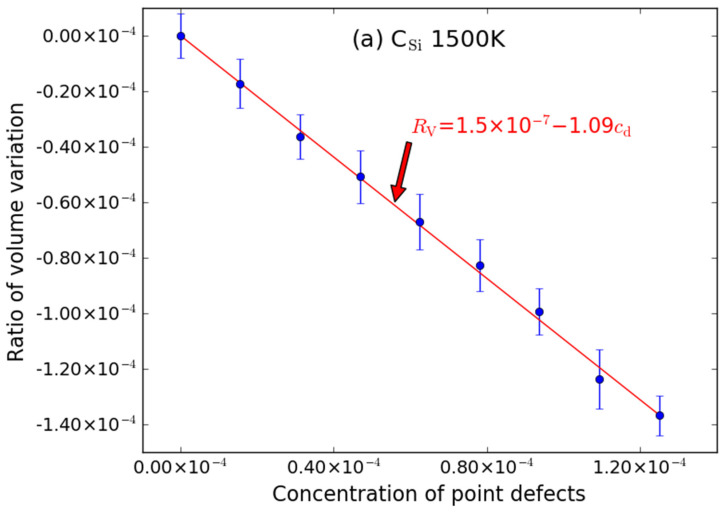
The ratio volume variation (*R*_V_) of the supercell caused by anti-site point defects ((**a**) C_Si_ and (**b**) Si_C_) as function of concentrations (*c*_d_) of point defects at 1500 K. The error bars denote the standard deviations of the values. The size of supercell in MD simulation is 20 *a* × 20 *a* × 20 *a*.

**Figure 8 materials-15-03008-f008:**
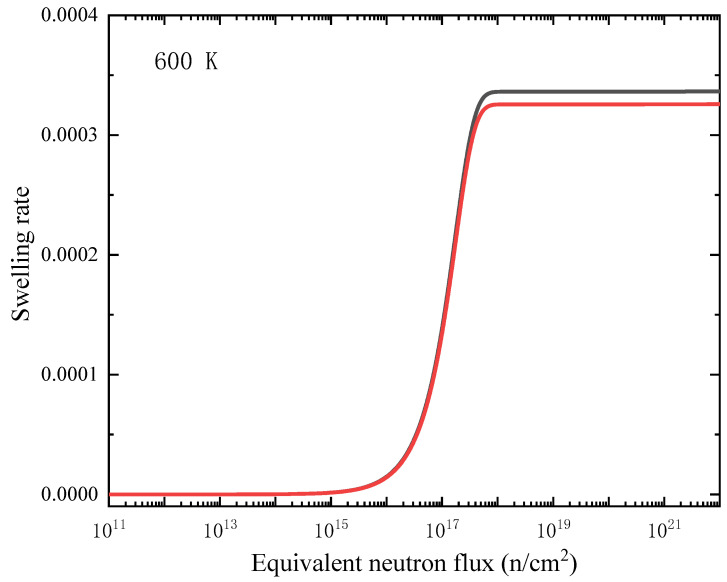
Relationship between swelling rate and equivalent neutron flux (1 MeV) in the central region of SiC film at 600 K. Black line corresponds the calculations with excess volumes from small-size (10 *a* × 10 *a* × 10 *a*) supercell simulations and the red line corresponds the calculations with excess volumes from large-size (20 *a* × 20 *a* × 20 *a*) supercell simulations.

**Figure 9 materials-15-03008-f009:**
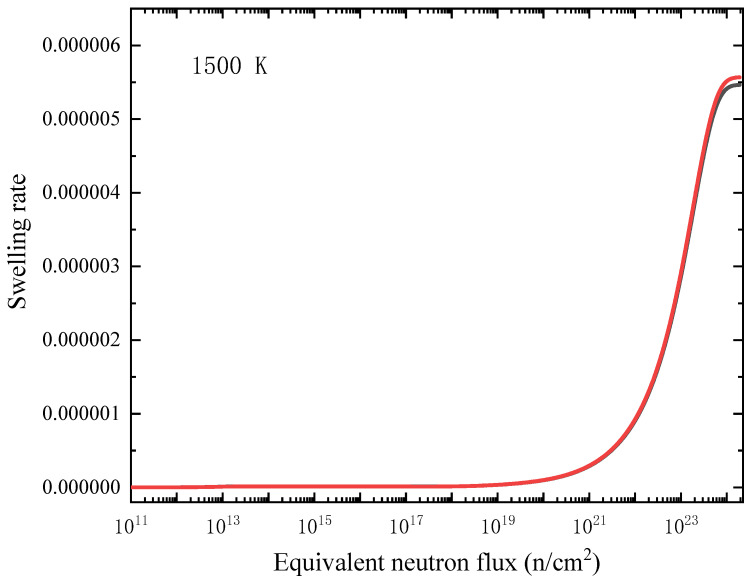
Relationship between swelling rate and equivalent neutron flux (1 MeV) in the central region of SiC film at 1500 K. Black line corresponds the calculations with excess volumes from small-size (10 *a* × 10 *a* × 10 *a*) supercell simulations and the red line corresponds the calculations with excess volumes from large-size (20 *a* × 20 *a* × 20 *a*) supercell simulations.

**Table 1 materials-15-03008-t001:** Excess volume of various point defects at 600 K and 1500 K (*ν_i_* in Å^3^). The values in parentheses are deduced from the small-size supercell (10 *a* × 10 *a* × 10 *a*) simulations.

Temperature/K	C_V_	Si_V_	C_I_	Si_I_	Si_C_	C_Si_
600	6.12 (5.12)	4.75 (4.13)	14.96 (16.64)	29.07 (27.55)	13.66 (13.68)	−11.49 (−11.79)
1500	4.95 (4.86)	4.61 (4.34)	16.04 (18.07)	30.51 (29.28)	14.50 (14.53)	−11.62 (−11.63)

## Data Availability

The data that support the findings of this study are available from the corresponding author upon request.

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
