# Peer review of "A Multi-Scale Simulation Study of Irradiation Swelling of Silicon Carbide"

_materials, 2022, doi:10.3390/ma15093008_

Round 1
Reviewer 1 Report
Reviewer’s comments on the manuscript: A multi-scale simulation study of irradiation swelling of silicon carbide written by Chunyu Yin, Baoqin Fu, Yongjun Jiao, Zhengang Duan, Lei Wu1, Yu Zou, Shichao Liu
The presented manuscript concerns the effect of neutron irradiation on the volume swelling of cubic-SiC film with 0.3 mm was studied by using the combination of molecular dynamics (MD) and rate theory (RT). It was found that for C-vacancy (CV), C-interstitial (CI), Si-vacancy (SiV), Si-interstitial (SiI) and Si-antisite (SiC), the volume of supercell increases linearly with the increase of concentration of these defects, while the volume of supercell decreases linearly with the increase of defect concentration for C-antisite (CSi).
The manuscript is in agreements with journal’s fields of interests. It is interesting and well organized. The obtained results are promising and clearly presented. Thus my suggestion is minor revision.
Reviewer's comments and suggestions:
- Abstract should underline the relevance of the presented studies.
- line 36: Please add some references.
- line 83: Please change the text into: “The research process includes the following steps. Firstly, MD….”
- All manuscript: please decade if you want to provide a space between the number and its unit or not, and correct all manuscript.
- line 193: please change the word “dependency” into “dependence”.
- lines: 222-229: Could you please rewrite this part of text to be more clear.
- References: There are a lot of editorial mistakes in the reference list. I advise you to check and correct them, item by item.
Reviewer 2 Report
Reviewer Comment for the authors:
This manuscript provides a study of the microscopic physical mechanism of silicon carbide (SiC) volume swelling induced by fast neutron irradiation using the combination of molecular dynamics (MD) and rate theory (RT)
The manuscript could potentially be suitable for publication after some minor revisions.
Please find the comments below:
- Since the role of point defects in volumetric swelling of SiC is nearly identical in the low and intermediate temperature regime. Please have a look to https://doi.org/10.1016/j.nimb.2015.04.059. How did the authors discuss this compared to their results?
- Page 3, line 105. Why does the authors choose these two temperatures (600 K and 1500 K) specifically? More explanation should be added.
- Page 2, line 67. Reference 16 should be replaced by the author's names instead of reference number only.
- Page 3, line 92. The term ZBL should be identified.
- I am extremely sympathetic to the challenges of writing in another language, but in the present case the language construction is not suitably clear as to be able to recommend publication. e. Page 3, line 94 the sentences started as follows “And the size of the cubic supercell is 20 a × 20 a × 20 a”. Therefore, the authors should check the manuscript very carefully and correct all possible typing and grammatical errors appeared in this manuscript. It’s highly recommended to send the manuscript for editing and/or proofreading service.
- The resolution of the figures should be improved. It is very important to add the highest possible resolution of the figure in the manuscript i.e. Fig. 1 and 5.
- References should be revised carefully as well as uniformly formatted. For example, please have a look to ref. 21, 27, 31, 35 and 36. These references have need to be revised compared to other references i.e. page number, year and journal name. Also, Ref. 18 is written wrongly.
